# Synergistic-potential engineering enables high-efficiency graphene photodetectors for near- to mid-infrared light

Hao Jiang[1,2,3], Jintao Fu[1], Jingxuan Wei [4], Shaojuan Li [5], Changbin Nie[1], Feiying Sun[1], Qing Yang Steve Wu [6], Mingxiu Liu[5], Zhaogang Dong [6] ✉, Xingzhan Wei [1] ✉, Weibo Gao [2] ✉ & Cheng-Wei Qiu [3] ✉

High quantum efficiency and wide-band detection capability are the major thrusts of infrared sensing technology. However, bulk materials with high efficiency have consistently encountered challenges in integration and operational complexity. Meanwhile, two-dimensional (2D) semimetal materials with unique zero-bandgap structures are constrained by the bottleneck of intrinsic quantum efficiency. Here, we report a near-mid infrared ultra-miniaturized graphene photodetector with configurable 2D potential well. The 2D potential well constructed by dielectric structures can spatially (laterally and vertically) produce a strong trapping force on the photogenerated carriers in graphene and inhibit their recombination, thereby improving the external quantum efficiency (EQE) and photogain of the device with wavelength-immunity, which enable a high responsivity of 0.2 A/W–38 A/W across a broad infrared detection band from 1.55 to 11 μm. Thereafter, a room-temperature detectivity approaching $1 \times 10^9$ cm Hz$^{1/2}$ W$^{-1}$ is obtained under blackbody radiation. Furthermore, a synergistic effect of electric and light field in the 2D potential well enables high-efficiency polarization-sensitive detection at tunable wavelengths. Our strategy opens up alternative possibilities for easy fabrication, high-performance and multifunctional infrared photodetectors.

Infrared sensing compatible with complementary metal–oxide–semiconductor (CMOS) technology has been the core of various optoelectronic applications, especially in the mid-infrared spectral range (multiple atmospheric windows of 3–5 μm and 8–14 μm for thermal imaging and environmental monitoring)[1–7]. Commercialized photodetectors for the mid-infrared range, such as HgCdTe and InAsSb/AlAsSb superlattices, face challenges related to compatibility with complementary metal–oxide–semiconductor (CMOS) technology, manufacturing complexity, and limitations associated with low-temperature operation. In contrast, graphene with broadband optical absorption capability and easy-integration features offers unique opportunities for on-chip infrared detection[8–15]. However, due to the limitation of internal quantum efficiency on the excitation and transport of incident photocarriers, the performance of pure graphene devices in most reports is currently not comparable to commercial photodetectors or image sensors[1]. A series of physical

[1]Chongqing Institute of Green and Intelligent Technology, Chinese Academy of Sciences, Chongqing, China. [2]School of Physical and Mathematical Sciences, Nanyang Technological University, Singapore, Singapore. [3]Department of Electrical and Computer Engineering, National University of Singapore, Singapore, Singapore. [4]School of Optoelectronic Science and Engineering, University of Electronic Science and Technology of China, Chengdu, China. [5]Changchun Institute of Optics, Fine Mechanics and Physics, Chinese Academy of Sciences, Changchun, China. [6]Institute of Materials Research and Engineering (IMRE), Agency for Science, Technology and Research (A*STAR), Singapore, Singapore. ✉e-mail: dongz@imre.a-star.edu.sg; weixingzhan@cigit.ac.cn; wbgao@ntu.edu.sg; chengwei.qiu@nus.edu.sg

mechanisms are attempted for developing the broadband detection capability of graphene, such as thin coherent absorption effect[16–22], photo-thermoelectric effect[23,24] and bolometric effect[25,26]. However, most still face limitations such as narrowband photoresponse, low responsivity, and low-temperature requirement (Fig. 1a).

A potential developmental trend in current infrared detection technology involves achieving high responsivity and integration compatibility with silicon-based systems at room temperature. This promises a heterogeneous platform capable of meeting the demands of future functional detection. Device architecture design with efficient signal acquisition and amplification functions are expected to be a highly competitive choice, including charge-injection effect[27], and manipulated photogating effect[28–36]. These methods can achieve high EQE from the perspective of charge transfer and amplification mechanisms to detect light with lower photon energy (*i.e.*, longer wavelengths). Therefore, to excite the wide-spectrum detection capability of graphene while ensuring high responsivity, an adaptive method is urgently needed to integrate monolithically with silicon-based integrated circuits in a back-end-of-line process.

Here, we introduce a 2D dielectric slit-induced potential well into the monolayer graphene photodetector with manipulated photogating mechanism, yielding broadband (from short-wave infrared to long-wave infrared (1.55–11 µm)) and high responsivity detection (0.2 A/W-38 A/W) at room temperature. Regarding the device structure,

graphene is wet-transferred onto dry-patterned silicon-on-insulator and connected to drain electrodes to serve as the channel (Fig. 1b). We found that the structural design of 2D dielectric slits can introduce periodic surface potential wells on the surface of graphene. This will spatially (laterally and vertically) produce a strong trapping force on the photogenerated carriers in graphene and inhibit their recombination, thereby improving EQE and photogain of the device. Because this design is based on the electrical gain derived from the carrier amplification mechanism, it enables an enhancement in responsivity across an ultra-wide spectral range. As a result, a room-temperature detectivity can reach up to nearly $1 \times 10^9$ cm Hz$^{1/2}$ W$^{-1}$ under blackbody radiation. Under the synergistic action of electric field and optical field, the specifically configured dielectric structure can be used not only as a trap material, but also as a metasurface structure with selectivity for polarized light. Not less importantly, this design is realized by an easy-integration graphene/silicon-based composite device. Therefore, it is compatible with CMOS technology and hold great potential application prospects for high-performance, multi-dimensional integrated mid-infrared detection.

## Results

### Design of graphene/2D slit structure photodetectors

The design mechanism of our device is elaborated in Fig. 1c. The generation of electrical gain encompasses three primary physical

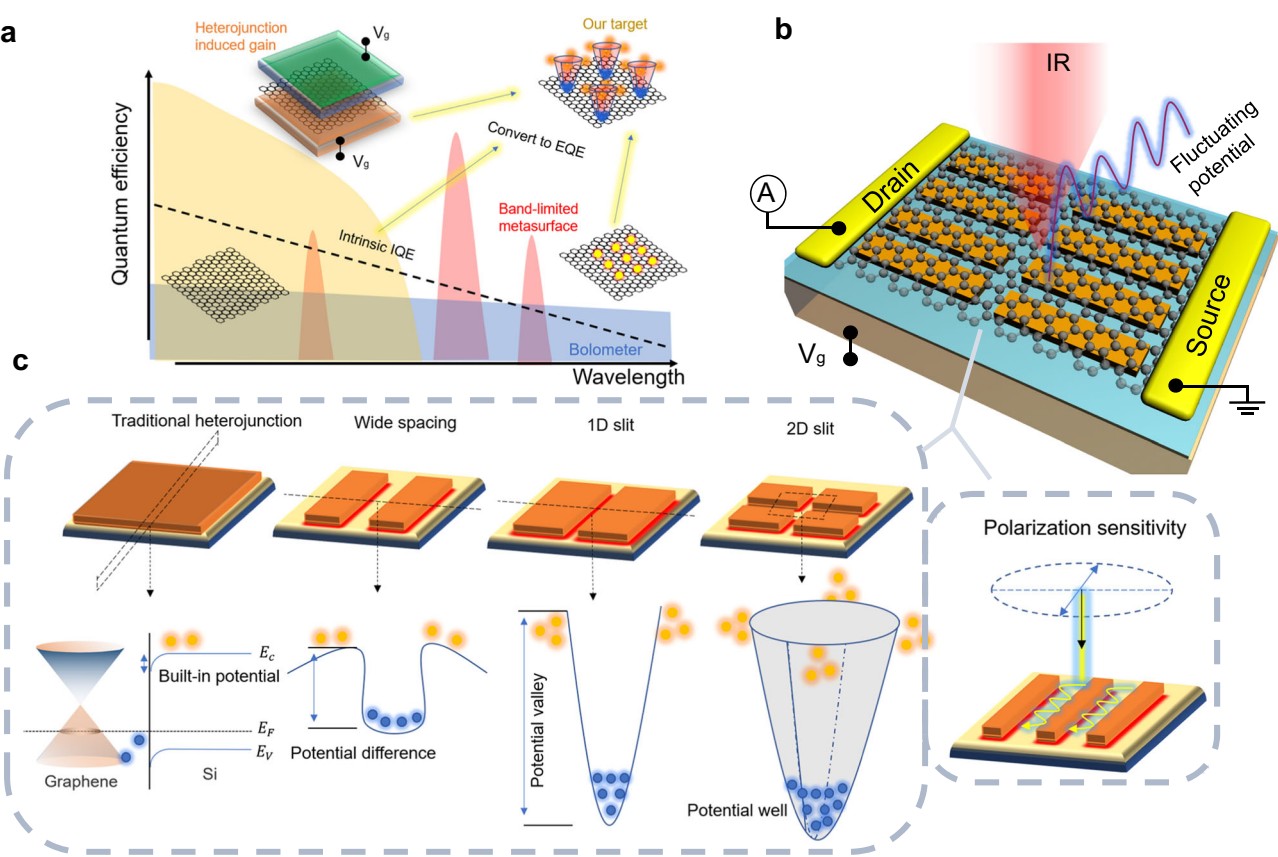

**Fig. 1 | Principle and mechanism. a** Comparison of different mechanisms for wide spectral detection of graphene. Limited by the internal quantum efficiency of graphene, high responsivity is hard to achieved solely through methods such as metal junctions or bolometer[14,25,26]. The assistance of metasurfaces can mostly only provide narrowband response[7,24]. Introducing heterostructures with gain effects can break through the bottleneck of internal quantum efficiency, but the spectral range remains limited[27–36]. A design approach is needed to simultaneously achieve high gain and wide-band detection targets. **b** Illustration of the designed graphene photodetector with configurable all-dielectric surface, which consists of silicon

gratings with different duty cycle (DC) and 2D slit structures. The fluctuating potential is thus introduced. **c** Schematic diagrams of interface potentials of different dielectric surface configurations and corresponding separation effects on photogenerated carriers. Polarization selectivity for linearly polarized light can be achieved by specific configuration of anisotropic grating structure. $E_C$, $E_V$, and $E_F$ respectively represent the conduction band, valence band, and Fermi level of the silicon band structure. The blue spheres represent electrons, and the orange spheres represent holes.

processes: the first entails the creation of potential wells, the second involves the separation of photocarriers, and the third encompasses the photoconductive gain process, as detailed in Supplementary Fig. 7 of Supplementary Note 2.

The separation of photogenerated carriers in traditional silicon-based graphene photodetectors mainly relies on the limited built-in electric field formed on the surface of graphene and silicon[29,30,37–39]. It has been discovered that a strong electric field exists at the edge of silicon and silicon oxide, surpassing the intensity of the built-in electric field formed by graphene and silicon[32,40,41]. Therefore, the periodic isolated nanograting can provide fluctuating surface potential as a driving force for separating photocarriers. That is, the amplified surface potential will trap more separated photo-induced charges of graphene into the silicon strip, leading to the enhanced photogating effect. Actually, the spacing and duty cycle (DC) of gratings play a crucial role in surface potential engineering. Under the condition of DC = 0.5, equidistant potential fluctuations will form on the dielectric surface, where a strong local electric field will be generated at the junction of silicon and silicon oxide. As the DC increases, the width of the low potential region will become narrower, forming a slit structure. Due to the electric field coupling between adjacent grating boundaries, the surface potential distribution will resemble a valley with a higher potential difference, as shown in the corresponding schematic diagram of Fig. 1c.

When light illuminates the device, photocarriers in graphene covering the surface will be separated under such a potential distribution, confining electrons to low potential regions while holes flow to high potential regions, thus intensifying the photogating effect to generate gain. A detailed analysis of the process can be seen in Supplementary Note 2. This potential distribution caused by the slit structure can be called the slit effect. Furthermore, when a slit structure is constructed in two mutually perpendicular directions in a two-dimensional plane, the valley potential distribution will evolve into a 2D potential well. At this point, the slit effect becomes a 2D slit effect, which will achieve the further enhancement effect. Besides, such a photoconductive gain induced by the distribution of electric field can also be coordinated with light field. For example, configurable dielectric structures can be specifically designed as anisotropic dielectric grating structures, serving as a metasurface sensitive to linearly polarized light. In general, these configurable dielectric surfaces can simultaneously control the interface electric field, trap regions, and polarization-sensitive characteristics, thereby realizing photodetection with multifunctional capabilities.

Further calculations can be conducted to analyze the surface electric field distribution of slit structures by TCAD, as shown in Fig. 2a. The top sub-graph shows the three-dimensional distribution of the electric field of the 2D slit structure. It can be clearly seen that the strong electric field at the edge of the stripes can couple with the weak electric field at the center of the groove to form a valley potential. Similarly, a potential well can be formed by surrounding hot spots at the intersection of valley potentials in both directions. Under the action of the 2D slit structure, the surface potential is distributed as displayed in the schematic diagram (bottom sub-graph). This inhomogeneous potential distribution will affect the separation and arrangement of photogenerated carriers. The corresponding electric field distribution for the cross-sectional and longitudinal sections of the 2D slit structure is illustrated in Fig. 2b. By intercepting the transverse electric field, the relationship between the electric field and the interface position can be obtained. We also calculated the electric field variation with position for DC = 1 (all silicon), DC = 0.5, and DC = 0.8, respectively, as illustrated in Fig. 2c. The surface potential of all-silicon dielectric remains constant, whereas, the periodic slit structure can lead to the generation of valley potential. As the gap spacing decreases, the coupling effect of the edge electric field becomes stronger, inducing larger local electric fields and deeper valley potentials. We measured the surface potential of the structure and obtained electric field distribution maps that were consistent with the experiment (See Supplementary Fig. 5 and Supplementary Fig. 6 in Supplementary note 2).

## Slit effect and characterization of device

The above slit effect can dramatically amplify the photocurrent from graphene by providing high photogain, as schematically shown in Fig. 3a. The slit structure formed by increasing the DC of the silicon grating (shortening the grating spacing) will evolve the enhanced built-in electric field on both sides of the grating into a valley potential distribution, which is manifested as a spatial potential well in the 2D slit structure. Under the well potential, photocarriers in graphene can be efficiently separated. In the high potential region, photo-induced holes are injected into silicon, while photo-induced electrons remain in graphene and are confined to the low potential region. That is to say, the enhanced built-in electric field here plays two driving roles:

$$qV_{bi} \propto \psi_s{}^{G-S} + \psi_s{}^{G-G} \tag{1}$$

One is the heterojunction electric field $\psi_s{}^{G-S}$ that separates photo-induced holes into silicon grating, and the other is the homojunction electric field $\psi_s{}^{G-G}$ that traps photo-induced electrons into potential wells. This strong separation and trapping effect can spatially enhance the photogating effect and improve quantum efficiency. The

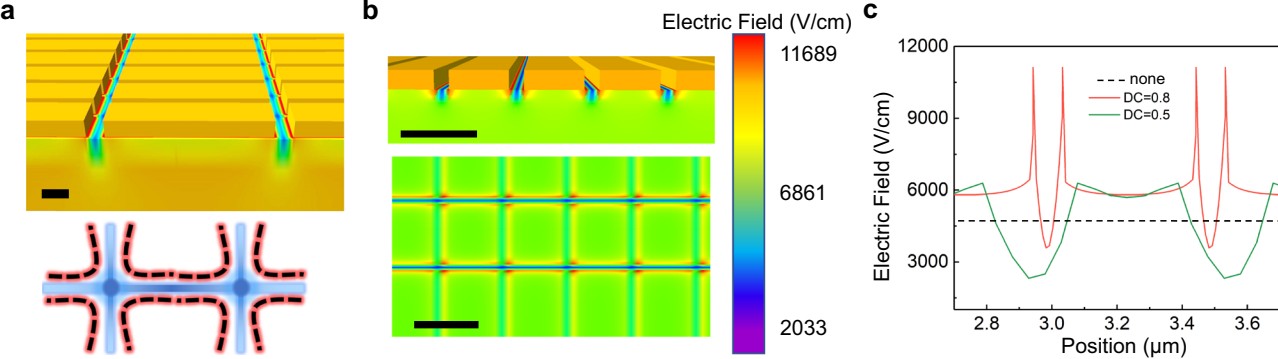

**Fig. 2 | Interface electric field analysis. a** Studies on the electric field distribution induced by slit structure. The top sub-figure shows 3D plot of the slit structure and the simulated field distribution, in which the blue area represents the low potential area, while the red area represents the high potential area, as bottom sub-figure. **b** The simulated potential distribution of the slit structure in cross-sectional and longitudinal sections. The grating height is 160 nm, the unit length is 1 μm, and the DC is 0.8. **c** Simulation of surface electric field distribution with lateral position at DC = 0.5, DC = 0.8, and no grating structure. Black scale bar in **a**, **b**, 1 μm.

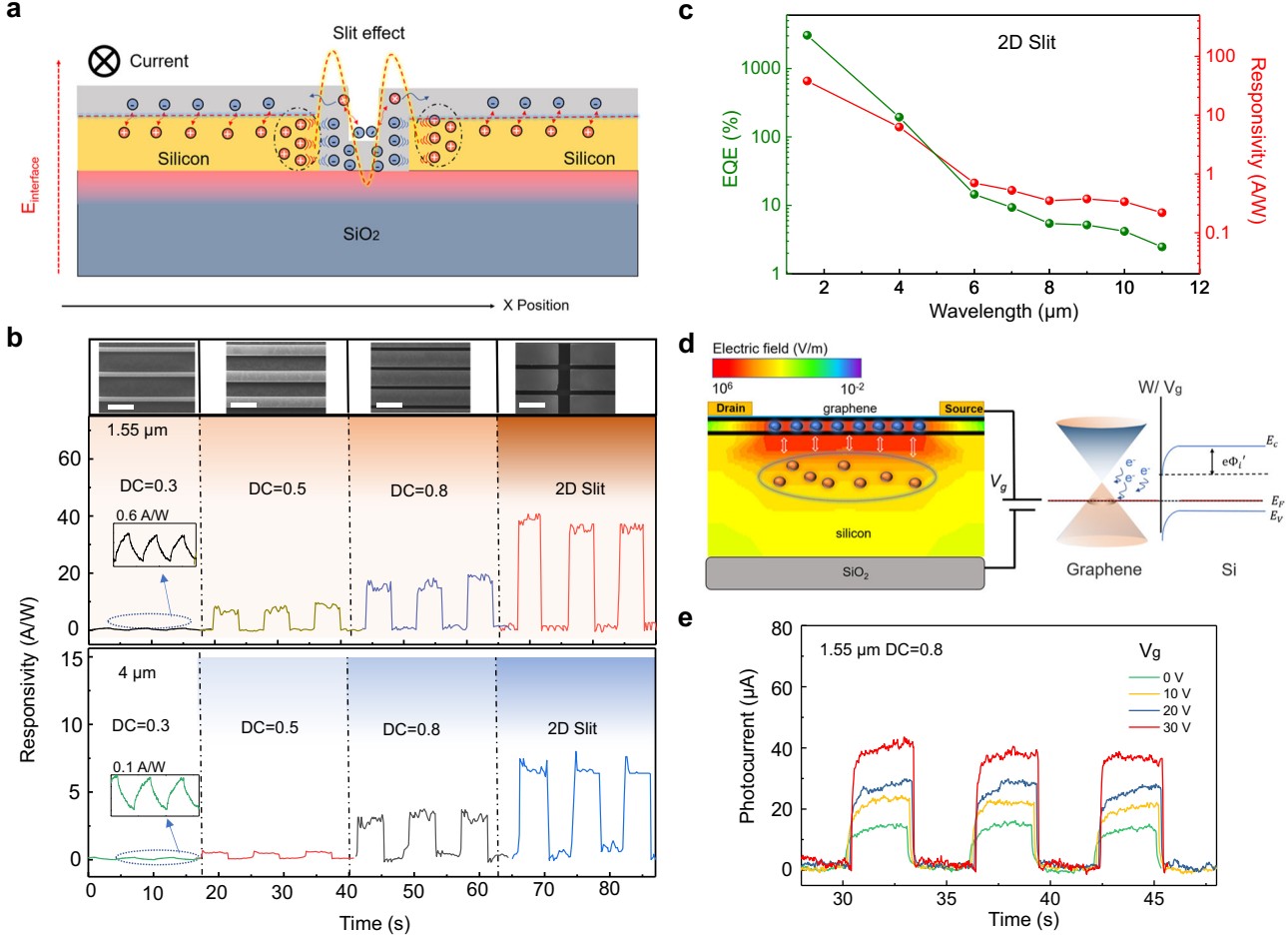

**Fig. 3 | Device performance and electrical test analysis. a** Schematic diagram of slit effect on the separation of photogenerated carriers in graphene. The potential well formed by the slit effect enhances the separation and capture of photo-generated carriers, leading to higher quantum efficiency and gain. The blue dashed line represents the interface electric field between silicon and graphene, while the yellow dashed line indicates the enhanced electric field caused by the slit. The red arrows denote the separation effect of both on the photocarriers. **b** The responsivity of different devices measured during on-off cycles of incident lasers at 1.55 μm and 4 μm wavelength. The illustration above shows a SEM image of the corresponding device structure. White scale bar is 1 μm. **c** Responsivity and EQE of graphene/2D slit structure device measured at different wavelengths. **d** Simulated schematic diagram of the manipulation of gate voltage on electric field distribution. The carrier injection effect caused by gate voltage can effectively enhance the interface electric field $e\Phi_i'$ by changing the Fermi level of graphene. The top-silicon height is 160 nm. The gate-voltage is 30 V. The blue spheres represent electrons, and the orange spheres represent holes. **e** The enhancement effect of gate voltage on the photoelectric response of devices with DC = 0.8. The wavelength of the incident light is 1.55 μm. The bias voltage of above tests and simulations is 0.1 V.

derivation is shown in Supplementary Note 2. According to the expression of gain as[29]

$$G = \eta(\tau_r, \mu) \cdot \omega(\psi_s(N, n_0)) \qquad (2)$$

Two figure-of-merits, namely the lifetime-gain factor $\eta(\tau_r, \mu)$ and the potential-gain factor $\omega(\psi_s(N, n_0))$, can be simultaneously improved here due to enhanced built-in electric field and trapping effect. The lifetime-gain factor is mainly determined by the recombination lifetime $\tau_r$ and the mobility of conduction carriers μ, and the potential-gain factor is mainly influenced by the carrier concentrations $N$ of silicon and $n_0$ of graphene. The estimation of device lifetime can be seen in the test of Supplementary Fig. 19.

We measured the infrared responsivities of devices with different DC, as shown in Fig. 3b. As the DC increases, the stronger photoelectric chopping signal can be obtained and reaches its maximum in the 2D slit structure due to electrical gain. Calculated peak responsivity can reach 40 A/W at 1.55 μm and 7.5 A/W at 4 μm. Furthermore, Fig. 3c shows the responsivity of the device measured by a quantum cascade laser and corresponding EQE calculated from the measured responsivity as $EQE = Rhc/q\lambda$, where q, h, λ, and c are

electron charge, Planck's constant, wavelength, and speed of light, respectively. The gain generated by the enhanced photogating effect can lead to high responsivity and EQE in the long-wave infrared region (More data can be seen in Supplementary Fig. 8 of Supplementary Note 2).

According to the testing and analysis, the 2D slit structure can create hotspots at the edges and intersections of the grating, greatly enhancing the separation efficiency of photocarriers. In fact, the separation efficiency of the photogenerated carriers in the graphene/ silicon interface, which is far away from the hotspots, can be further improved by synergistic effect of back-gate voltage. When a positive back-gate voltage is applied, a carrier accumulation effect is formed in the semiconductor material, leading the injection of electrons into graphene. This can increase the built-in electric field at the interface, thereby improving the overall quantum efficiency, as shown in Fig. 3d[41–45]. The corresponding energy band diagram is shown in right sub-graph of Fig. 3d. The carrier injection effect caused by gate voltage can effectively enhance the interface electric field by changing the Fermi level of graphene. Applying gate voltage modulation to devices with the DC of 0.8 caused an improvement in the response signal to light of 1.55 μm wavelength, as shown in Fig. 3e.

This indicates that both the slit structure and back-gate voltage can improve the device responsivity by enhancing the interface electric field. This observation also helps clarify whether the interface electric field can effectively affect the photogating mechanism for such structures (More details can be seen in Supplementary Figs. 9 and 10 of Supplementary Note 2).

## Polarization sensitive detection

The configurable dielectric structure provides us with a platform that can be flexibly designed according to our requirements to achieve functional detection with high responsivities, such as polarization sensitive detection. Currently, on-chip integrated polarization-sensitive detectors mainly rely on anisotropic two-dimensional materials or metasurface structures[7,24,46–49]. Anisotropic metal metasurface structures can compensate for the low polarization ratio of anisotropic materials. However, most face the problem of low responsivity. Here, the gain induced by electrical field engineering can synergistically interact with the anisotropic distribution of light field. By designing the dielectric material that can serve as photogating trap material into an anisotropic structure, we can effectively achieve polarization sensitive detection with high responsivity. With the design parameters set to DC = 0.3, H = 160 nm, and L = 1 μm, the structure exhibits a high reflection polarization ratio for light of 1.55 μm in the X (perpendicular to the grating) and Y (parallel to the grating) polarization directions, as shown in Fig. 4a. The simulation (dots) and experimental measurements (lines) of far-field characteristics are highly consistent (See Supplementary Fig. 11 for different DCs). We can refer to the points with the strongest and weakest reflections as **R** point and **A** point. The electric field distribution caused by the incidence of polarized light inside the grating exhibits significant differences. From the near-field characteristics of the light field intensity distribution corresponding to the incident light at 1.55 μm (Fig. 4b), it can be seen that Y-polarized light is largely reflected (**R** point). In contrast, X-polarized light is localized around the silicon strip (**A** point), leading a difference in absorption of graphene, as shown in Supplementary Fig. 12. Supplementary Fig. 13 in Supplementary Note 3 shows the near-field distribution under different polarization angles of light. Hence it will have an impact on the number of photogenerated carriers participating in the photogating process, further leading to varying degrees of gain. The measured chopping signals with different polarization angles are shown in Fig. 4c. The linear polarization state of the incident light at 1.55 μm is controlled via rotation of the half-wave plate (HWP). It can be observed that as the polarization angle of the incident light changes, the near-field enhancement of the structure and the measured photocurrent show a high degree of correlation, including the corresponding "R" and "A" points, as shown in Fig. 4d. Through multiple measurements, a sinusoidal function relationship between the photocurrent and the polarization angle can be obtained, as shown in Fig. 4e. The maximum polarization ratio is 10, and the responsivity is about 1.43 A/W. In addition, by adjusting the structural DC to 0.8, polarization sensitive detection can be achieved at wavelength of 4 μm, as shown in Fig. 4f (See Supplementary Fig. 14). Nevertheless, in the case of the 2D slit structure, no particular resonance mode exists for the test wavelength range, making it suitable for emphasizing its remarkably high responsivity characteristics. Based on these structure, more flexible designs can be achieved, for example, by modulating the conductivity of graphene through gate voltage, one can tailor polarization ratio and wavelength, as shown in Supplementary Fig. 15 of Supplementary Note 3.

## Blackbody characterization of device

Blackbody detection, as a detection standard for practical applications, is used to demonstrate the infrared detection performance of photodetectors. Further, we investigated the devices for blackbody detection at room temperature. Figure 5a presents the schematic diagram of the blackbody detection system. The blackbody source is Cisystems SR200N with an adjustable temperature from 500 K to 1200 K. The device was placed in front of the aperture with a fixed modulation frequency of chopped by an optical chopper wheel. Here, the calculation of real black-body radiation power on device $P$ needs to take into account the blackbody temperature $T$, background temperature $T_0$, aperture radius of the blackbody radiation source $r$, the distance between the aperture and the detector $d$, the detector photosensitive area $A$ and black-body radiation rate $E_r$. According to the Stefan-Boltzmann law of $E_r = \frac{2\pi^5 k_B^4}{15 h^3 c^2} \cdot (T^4 - T_0^4)$[50–52], where $c$ is the speed of light, $h$ is the Planck constant, and $k_B$ is the Boltzmann constant. Thus, $P$ can be calculated by $P = \frac{E_r \cdot r^2 \cdot A}{d^2}$. The relationship between blackbody emission power without background temperature is shown in Supplementary Fig. 16 of Supplementary Note 4. Under black-body radiation at different temperatures, the device can display stable photoelectric chopper signals, as shown in Fig. 5b. Furthermore, the relationship between the measured responsivity and the equivalent noise power (*NEP*) with temperature under blackbody radiation was obtained, as shown in Fig. 5c, d.

*NEP* is calculated by $NEP = \frac{i_n}{R_p} = \frac{\sqrt{S(f_n)}}{R_p}$[53], where $S(f_n)$ is the real-time measured current noise power spectral density, $f_n$ is the center frequency of the device, and $R_p$ is the responsivity. Under the effect of photogating gain, the devices generally exhibit high responsivity and low *NEP*. Among them, 2D slit structure devices exhibit superior performance, which is attributed to the enhancement of the photogating effect by the slit effect.

The detectivity approaching $1 \times 10^9$ cm Hz$^{1/2}$ W$^{-1}$ can be obtained by $D^* = \frac{\sqrt{A \cdot \Delta f}}{NEP}$, where $\Delta f$ represents the working bandwidth (1 Hz), and $A$ is the device area. During the testing process, there was no significant change in the device's resistance, thus the interference caused by photothermal effects can be neglected (See Supplementary Note 4 for more device performance and noise data under black-body radiation in Supplementary Figs. 17 and 18).

## Discussion

The graphene/2D slit structure photodetector based on the slit effect demonstrated here is critical for many technologies that require high sensitivity such as safety surveillance, object inspection, and astronomical observation[3,54–57]. Besides, our devices operate at room temperature and have a broadband operating range, which is highly desired for the next generation infrared photodetectors[58]. Importantly, our work provides configurable design for multi-functional optoelectronic devices. For example, the device can realize linear polarization sensitive detection with high gain, and we expect that the design of chiral structure can further realize circular polarization sensitive detection in the future. This work shows advantages in detection wavelength and responsivity compared to other graphene infrared detectors[59–66], as shown in Supplementary Table 3 of Supplementary Note 5. Moreover, the fabrication of our devices is fully compatible with the silicon-based CMOS process, which is of paramount importance for the realization of miniaturized high-gain infrared detectors.

## Methods

### Device fabrication

For the fabrication of the structured substrate, the graphene/silicon-on-insulator (SOI) substrates with 160 nm heavily-doped top-silicon and 300 nm oxide layer were chosen. Firstly, polymethylmethacrylate (PMMA) was used as photoresist to write different patterns on the top-silicon surface through electron-beam lithography (EBL). Then, a 20 nm chromium layer was evaporated on the surface. After liftoff process, the patterned chromium was left as a masking layer. Next, the bare silicon region was etched by dry etching process with

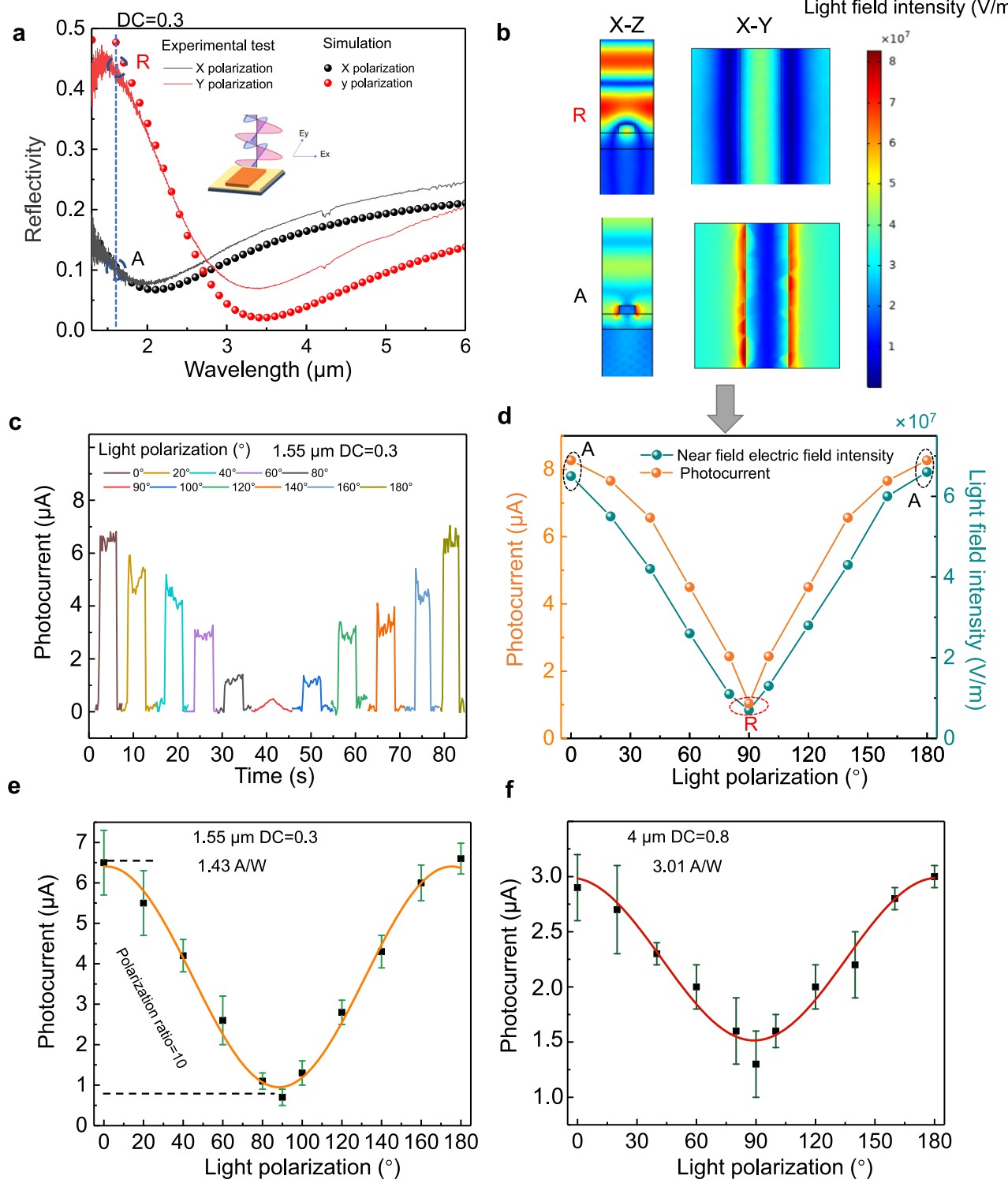

**Fig. 4 | Polarization response of device. a** Measured reflective polarization sensitivity characteristics by Fourier Transform Infrared Spectroscopy (FTIR) and simulation of structured devices with DC of 0.3. The grating height is 160 nm, the unit length is 1 μm. The grating structure has strong linear polarization sensitivity near 1.55 μm with **A** point and **R** point. **b** Near field simulation of this structure for illumination along different polarization directions (**A** point and **R** point). **c** Chopped photocurrent signals obtained by changing polarization angles of incident light. (λ = 1.55 μm). **d** The relationship between near-field enhancement of different polarized light and measured photocurrent. **e, f** The relationship between the magnitude of the photocurrent measured multiple times and the polarization angle of the incident light at wavelength of 1.55 μm and 4 μm, respectively. The bias voltages in all tests are 0.1 V. The error bars represent the range of photocurrent values obtained from multiple measurements. The two dashed lines are used to scale the difference in photocurrent caused by polarization for calculating the polarization ratio.

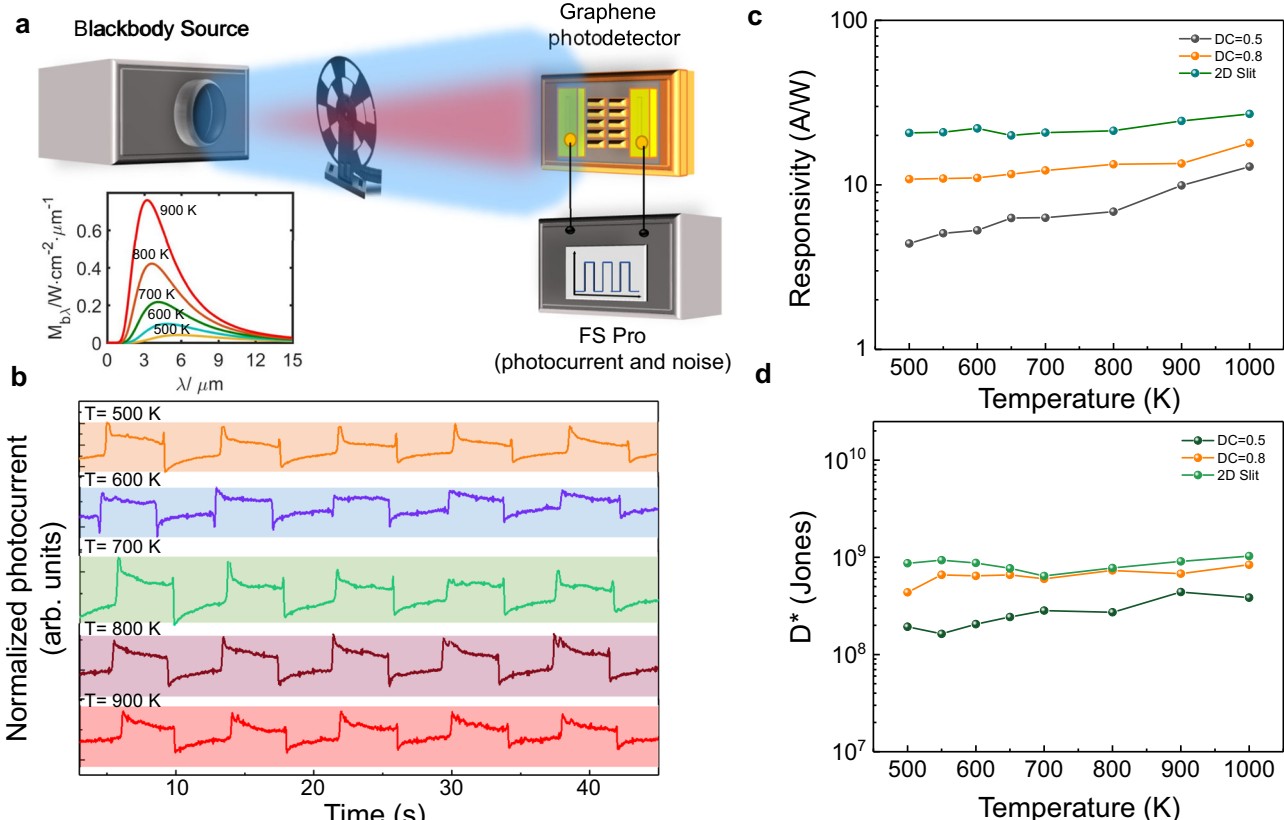

**Fig. 5 | Blackbody Characterization of device. a** Blackbody measurement schematic of graphene/nanostructured all-dielectric photodetectors under blackbody source illumination. The illustration depicts the Planck formula curves for different color temperatures, calculated based on the blackbody radiation source we utilized. It shows the corresponding monochromatic radiance for different color temperatures and wavelengths, according to the inference of Supplementary Note 4. **b** Photoelectric chopping signal of 2D slit device under black-body radiation at different temperatures. **c**, **d** The relationship between the responsivity and specific detectivity (D*) of devices with different configuration structures as a function of blackbody temperature. The bias voltage is 0.1 V.

trifluoromethane and sulfur hexafluoride. Finally, chromium was dissolved to obtain the target substrate.

For the transfer of graphene, large scale graphene film was synthesized by the chemical vapor deposition on copper foil. Subsequently, the Cu foil coated with graphene was spin-coated with PMMA as protective layer and baked on a hot plate at 150 °C for 10 min. A mixed solution of hydrochloric acid and hydrogen peroxide was used as an etchant to remove the copper film. Graphene coated with PMMA was dried and transferred to the target substrate. At final, the PMMA film was dissolved with acetone.

For the fabrication of electrodes, bi-layer photoresist, LOR and S1805, were spin-coated on the surface of graphene. Then, the samples with electrode patterns were obtained through binary exposure and development. Cr film (3 nm) and Au film (50 nm) were sputtered on the surface of graphene by magnetron sputtering. Finally, electrodes distributed on both sides of the dielectric structure were obtained by dissolving the unexposed photoresist with acetone solution. After the wet transfer of graphene onto a structured substrate, annealing was performed to ensure the conformal integration of graphene with the surface structure.

For the patterning of graphene thin films, A bi-layer (LOR and S1805) photolithography process was used to pattern the graphene film. After development, the uncovered graphene was etched by using oxygen plasma for about 30 s. Lastly, the graphene channels were obtained by dissolving the remaining photoresist in AZ400 solution, which conformally cover the dielectric structure and connect with the electrodes.

Scanning electron microscopy (SEM), atomic force microscope (AFM), and optical microscope images of structures can be seen in

Supplementary Fig. 1 and Supplementary Fig. 2 of Supplementary Note 1. The fabrication process of devices can be seen in Supplementary Fig. 3 of Supplementary Note 1. Parameters of the graphene can be seen in Supplementary Table 1 and Supplementary Fig. 4 of Supplementary Note 1.

## Characterization methods

1.55 μm laser, 4 μm laser, 6–12 μm quantum cascade laser were used as light sources for different wavelength tests in this work. Low-order half-wave plates designed at 1.55 μm and 4 μm were used to control the polarization angles of light. FS Pro Integrated Semiconductor Parameter Testing System (Noise Analyzer) was used to measure the photocurrent, noise, and resistance of devices. The polarization reflection spectrum of the grating structure was measured by FTIR with a polarizer. The surface potential of the structure was measured by AFM combined with a potential testing system. Cisystems SR200N was used as the black-body radiation source, and the maximum temperature can reach 1200 K. The setup of the device testing system is shown in Supplementary Fig. 20 of Supplementary Note 6. The detailed testing parameters for device noise and responsivity are shown in Supplementary Tables 2 and 4.

## Simulation

The numerical simulations of polarization sensitivity analysis in this work were carried out using the COMSOL. To Simulate the near-field and far-field characteristics of grating structures, we applied periodic boundary conditions at the x- and y-boundaries and perfect-matched layer (PML) conditions at the z-boundaries.

Silvaco TCAD was used to conduct three-dimensional simulation of the distribution of electric field and charge carriers at the device interface.

## Data availability

All technical details for producing the figures are enclosed in the supplementary information. Data are available from the corresponding authors C.-W.Q. or X.W. upon request.

## Code availability

All technical details for implementing the simulation are enclosed in the Supplementary Information. Codes are available from the corresponding authors C.-W.Q. or X.W. upon request.

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

## Acknowledgements

This work was supported by the National Key R&D Program of China (2017YFE0131900), the Natural Science Foundation of Chongqing, China (cstc2019jcyjjqX0017), the National Research Foundation, Singapore, and A*STAR under its Quantum Engineering Programme (NRF2021-QEP2-03-P10), National Natural Science Foundation of China (Nos. 62121005, 62022081 and 61974099), Changchun Key Research and Development Program (21ZY03). Z.D. would like to acknowledge the funding support from Agency for Science, Technology and Research (A*STAR) under its AME IRG (Project No. A20E5c0093), Career Development Award grant (Pro-ject No. C210112019), MTC IRG (Project Nos. M21K2c0116 & M22K2c0088) and Quantum Engineering Programme 2.0 (Award No. NRF2021-QEP2-03-P09). C.-W.Q. acknowledges finan-cial support from the NRF, Prime Minister's Office, Singapore under the Competitive Research Program Award (NRF-CRP26-2021-0063).

## Author contributions

H.J., X.W. and C.-W.Q. conceived the project. H.J., J.F., Q.W. and J.W. performed the measurements. H.J., J.F. and Z.D. fabricated the devices. H.J. and J.W. analyzed the data. H.J. performed the theo-retical analysis. H.J., C.-W.Q., S.L., X.W. and W.-b.G. wrote the manuscript. X.W., C.-W.Q. and W.-b.G. supervised the project. All the authors, including C.N., F.S. and M.L. contributed to the dis-cussion of the results.

## Competing interests

The authors declare no competing interests.
