## [Peer Review File · Nature Communications]

Synergistic-potential engineering enables high-efficiency graphene photodetectors for near- to mid-infrared lightREVIEWER COMMENTS

Reviewer #1 (Remarks to the Author):

In this manuscript, Qiu et al. show a very interesting example of near- to mid-infrared photodetector based on graphene. I appreciated very much reading their manuscript and I recommend its publication in Nature Communications. I would only ask the authors to address the following points:

- I understand that all the fabrication steps have been moved to the methods section, however, while reading the text, it is difficult to understand how the heterostructure of their detector is formed. I suggest summarizing in a brief sentence what device they propose (e.g. graphene wet-transferred on dry-patterned silicon-on-insulator [...]) before diving into the device characterization.
- I suspect that graphene is suspended when transferred on such a narrow slit. Have the authors taken this into account in their device analysis?
- Please double-check acronyms: R is never introduced as responsivity (see line 202) and HWP is never defined.
- The authors use a device which is in a photoconductive configuration. This means a higher dark current compared to photodiodes, photo-thermoelectric devices, etc. It would be nice if the authors could briefly comment on this.
- I find the valley and peak nomenclature a bit counterintuitive as, the most important parameter, i.e., the photocurrent/responsivity/detectivity, is, in fact, higher in the valleys and smaller in the peaks.

Reviewer #2 (Remarks to the Author):

The manuscript presents an interesting implementation of a layered material-based device where the electrical gain and light field tuning effects are combined in order to increase the responsivity and to extend the working range toward the long(er) wavelength.

There are significant noteworthy results, especially looking at the high responsivity reported for the longer wavelengths (measured with a blackbody radiation source).

The work is sufficiently novel, however, there are reported devices which recall the structure demonstrated here.

There is a substantial lack of information in the main article but especially in the supporting material(see below for details).

The soundness of the methodology is difficult to assess because of the lack of information mentioned before.

Because of the previous points, the fabrication of the device can be replicated, while replicating the actual results and relative tests will be not straightforward.

Here is a list of specific comments which I believe have to be addressed.

- 1) There are too many qualitative adjectives that, in this scientific context, don't carry valuable information, prevent clear understanding and repeatability and benchmarking.
- 2) Most of the fundamental physics behind the device's working principle is only described qualitatively without sufficient modelling/simulation/references.
- 3) Figure 1d the simulation seems to show an asymmetric behaviour in an otherwise (apparently) symmetric structure
- 5) Most of the figures and plots are not fully readable and in low-quality
- 6) The responsivity mentioned is expressed without any further information on the characterization conditions: applied source-drain voltage, condition of the illumination, spot shape and size, etc. This info are also missing in the supplementary material
- 7) The caption of figure 2 starts by saying "simulation...." But the overall picture does not include mainly simulation.
- 8) Blackbody radiation measurements have several conceptual gaps. To mention a few: how are the other effects that can induce a variation in the graphene resistance taken into account (thermal, strain, etc)?
- 9) Supplementary material: the "Surface Electric Field Engineering and Gain model" is mainly a standard model for pn junction in and out of equilibrium. The authors should explicitly adapt the model to the actual device under test.
- 10) No measurements of the graphene have been carried out. Raman, doping, strain and resistivity should be carried out. If some of them are not possible, at least proper characterization of the resistivity should be done.
- 11) Supplementary information: section "Transient photoresponses of graphene/2D slit structure photodetector upon different wavelength". The chopper speed is properly set to show the transient. Hence this can't be properly observed/quantified.
- 12) Supplementary material: in Table S2, several working principles are compared together: this comparison requires supporting evidence and explanations in order to give the tools to the reader to understand the quantities.

I believe that in its current status, the paper should not be accepted.

Reviewer #3 (Remarks to the Author):

This manuscript titled "Synergistic-Potential Engineering Enables High-Efficiency Graphene Photodetector for Near to Mid-infrared light" by Jiang et al reported the design and fabrication of graphene infrared photodetector based on 2D silicon-on-insulator substrate. The 2D potential well created by the patterned silicon block matrix can effectively trap photoexcited carriers and enable high

photoconductive gain. In addition, by designing the silicon matrix for polarized detection, the detector showed highly polarized photo response for 1.55 μm incident light. Finally, the detector was tested for detecting blackbody radiation from 500 K to 1000 K with high responsivity and detectivity. This manuscript provided a new design for enhanced infrared photodetection of graphene photodetector. The patterned silicon substrate is compatible with semiconductor processing technology, and the design can be tailored for various bands. Therefore, this manuscript is of great important for graphene infrared photodetectors. I would recommend its publication after the following comments are properly addressed in the revised manuscript.

1. The trapping of carriers by the potential well is crucial for the high responsivity, and the recombination lifetime is an important parameter. Can the authors measure or estimate the prolonged the carrier lifetime compared with conventional graphene/Si junction?
2. The detector showed responsivity up to 38 A/W. Is this high responsivity mainly from the long carrier lifetime or enhanced absorption? Did the authors measure the absorption of the device?
3. To achieve polarized detection, the silicon needs to be patterned for polarization sensitive response, however, the responsivity is much lower than that shown in Figure 2. It seems the 2D potential well and the polarized detection can not be attained simultaneously. Please comment on this.
4. For blackbody detection, the responsivity increases with temperature. Since the radiation power also increases with temperature, this trend is opposite to the normally observed decreasing responsivity with incident power. Did the author measured power dependent responsivity for 1.55 μm or other wavelength?
5. Table S2 is not complete.
6. What was the source-drain voltage and the corresponding electric field?

Reply to the reviewers:

Reviewer 1

Comments:

In this manuscript, Qiu et al. show a very interesting example of near- to mid-infrared photodetector based on graphene. I appreciated very much reading their manuscript and I recommend its publication in Nature Communications.

Author Reply: We appreciate the reviewer's positive feedback on this work and the valuable suggestions provided. We have made revisions and additions as per each point, including the inclusion of necessary figures and tables, as detailed below.

1. I understand that all the fabrication steps have been moved to the methods section, however, while reading the text, it is difficult to understand how the heterostructure of their detector is formed. I suggest summarizing in a brief sentence what device they propose (e.g. graphene wet-transferred on dry-patterned silicon-on-insulator [...]) before diving into the device characterization.

Author Reply: We appreciate the suggestions made by the reviewer. In the main text, there was indeed a lack of extensive elaboration on the device fabrication process, which left readers with a sense of information deficit. Consequently, we have supplemented this deficiency with detailed process flowcharts for device fabrication as shown in Figure R1.1. To summarize in short sentences: Graphene is wet-transferred onto dry-patterned silicon-on-insulator and connected to the source and drain electrodes to serve as the channel.

Author action: We have added an explanation “Regarding the device structure, graphene is wet-transferred onto dry-patterned silicon-on-insulator and connected to source and drain electrodes to serve as the channel (Fig 1b).” about the device structure on page 3 of our manuscript. Additionally, we have included Figure R1.1 in the supplementary information as Figure S3, along with an accompanying image description. The detailed description of the fabrication process can be found in **Method**.

Figure R1.1 Fabrication process of devices.

2. I suspect that graphene is suspended when transferred on such a narrow slit. Have the authors taken this into account in their device analysis?

Author Reply: In the experimental process, we also utilized AFM characterization to compare and determine the state of graphene on the structural surface. In this context, in the grating structure with the maximum duty cycle, when the grating height is 160 nm with a 200 nm spacing, graphene is less likely to be suspended. Furthermore, after wet transferring graphene onto the substrate structure, two annealing steps were performed, with the second annealing being particularly crucial. The 12-hour annealing process at 300°C had been conducted to ensure that graphene makes full contact with the substrate structure. AFM characterization was conducted to confirm the conformation of graphene with the substrate, as shown in the Figure R1.2, it can be observed that graphene approached a nearly conformal relationship with the grating structure.

Figure R1.2 Optical microscope image and AFM image of the surface structure covered with graphene when the grating height is 160 nm

Author action: We have added the description “After the wet transfer of graphene onto a structured substrate, annealing is performed to ensure the conformal integration of graphene with the surface structure” on Method section of the manuscript.

3. Please double-check acronyms: *R* is never introduced as responsivity (see line 202) and *HWP* is never defined.

Author Reply: We appreciate the issues raised by the reviewer. We apologize for not providing the full names of these specific terms promptly. We have defined the responsivity (*R*) and the half-wave plate (*HWP*), and conducted a comprehensive review and correction of all specialized terminology throughout the manuscript.

Author action: We have corrected the issues pointed out by the reviewers. We have re-examined the definitions of specialized terms throughout the entire manuscript.

4. The authors use a device which is in a photoconductive configuration. This means a higher dark current compared to photodiodes, photo-thermoelectric devices, etc. It would be nice if the authors could briefly comment on this.

Author Reply: We appreciate the questions raised by the reviewer. The device type in this paper is a photoconductive device based on the photogating effect, distinct from photodiodes and photo-thermoelectric devices, each of these three device types has its own advantages and disadvantages. Photodiodes and photo-thermoelectric devices utilize the separation and flow of photocarriers under asymmetric potential or temperature gradients, allowing them to operate at zero bias voltage, with low dark current and a high ON/OFF ratio. However, they lack gain and exhibit low

responsivity as disadvantages. Photoconductive devices based on the photogating effect primarily rely on the separation and recombination of charge carriers in vertical heterojunctions or traps to generate a gain, necessitating operation under bias voltage, hence achieving high responsivity. This, however, leads to the challenge of elevated dark current, which requires addressing in future work. In this paper, the non-uniform distribution of heterojunctions to some extent helps suppress the dark current in the graphene channel. These discussions indeed need to be summarized at the end of the paper.

As the reviewer has pointed out, different types of devices need to be discussed and compared separately, as shown in the table below. All these devices depend on graphene for light absorption, making photoconductive devices with gain advantageous for broad-spectrum detection. Our research harnesses the synergistic interaction between optical and electrical fields, enabling us to surpass the performance advantages documented in existing literature.

Table R1 Summary of device parameters of several typical graphene/semiconductor photodetectors previously reported, and our own device.

Types of devices	Working mechanism	Responsivity	Wavelength
Graphene/Si	Photodiode	0.28 A/W	1550 nm
TPA-doped tri-layer graphene/Si	Photodiode	0.435 A/W	800 nm
Graphene/CNT/SiO ₂ /Si	Photodiode	0.21 A/W	980 nm
Graphene–silicon-on-insulator	Photodiode	0.029 A/W	980 nm
Graphene double-layer heterostructure	photo-thermoelectric	1.1 A/W	3200 nm
Graphene–silicon heterojunction in conductor mode	Photoconductor	0.23 A/W	1550 nm
PtNPs/graphene/Si	Photoconductor	26 A/W	790 nm
MoTe ₂ /Graphene	Photoconductor	60 A/W	1064 nm
Au NP array/graphene	Photoconductor	83 A/W	1550nm
Graphene/ WS ₂	Photoconductor	0.735 A/W	1550nm
Graphene/silicon grating	Photoconductor	25 A/W	2700nm
This work	Photogating	0.2 A/W-38 A/W	1500-11000 nm

Author action: We have updated Table S4 in the supplementary information and added relevant explanations.

5. I find the valley and peak nomenclature a bit counterintuitive as, the most important parameter, i.e., the photocurrent/responsivity/detectivity, is, in fact, higher in the valleys and smaller in the peaks.

Author Reply: Sorry for the confusion caused by the previous definitions. As pointed out by the reviewer, the names of peak and valley are not appropriate. After careful consideration, we can refer to the points with the strongest and weakest reflections as **R** point and **A** point., as shown in Figure R1.4.

Author action: We have redrawn Figure 3 in the manuscript and revised the relevant instructions.

Figure R1.3 Revised Figure 3

Reviewer 2

Comments: The manuscript presents an interesting implementation of a layered material-based device where the electrical gain and light field tuning effects are combined in order to increase the responsivity and to extend the working range toward the long(er) wavelength.

There are significant noteworthy results, especially looking at the high responsivity reported for the longer wavelengths (measured with a blackbody radiation source).

The work is sufficiently novel, however, there are reported devices which recall the structure demonstrated here.

There is a substantial lack of information in the main article but especially in the supporting material (see below for details).

The soundness of the methodology is difficult to assess because of the lack of information mentioned before.

Reply: We greatly appreciate the reviewer's valuable questions, which highlight certain shortcomings in the depth of analysis and completeness of information in the paper. This has inspired us to make comprehensive revisions to the figures, tables, expressions, theoretical models, and missing information in the manuscript. In response to the reviewer's queries, in the revised manuscript, we have systematically reorganized the physical processes underlying the device operation. Additionally, we have provided comprehensive details on device simulation, fabrication, and testing, while also enhancing the clarity and refinement of the figures and corresponding descriptions throughout the entire document. We believe that these improvements, compared to the previous version, have significantly enhanced the paper, and we are thankful for the reviewer's insightful comments. Below are the itemized modifications addressing the reviewer's questions.

Comments 1: There are too many qualitative adjectives that, in this scientific context, don't carry valuable information, prevent clear understanding and repeatability and benchmarking.

Reply 1: We highly appreciate the invaluable questions raised by the reviewer. We have undertaken a thorough reevaluation and understanding of the phenomena and underlying mechanisms in our work, and have made revisions in the following areas to enhance the quality of the manuscript for clear understanding and repeatability and benchmarking:

1. We have undertaken a rephrasing of some of the qualitative descriptions in the mechanism explanation of Figure 1, and have categorized the associated physical processes into three distinct stages, as elaborated in detail in in Supplementary

Note 2.

2. For the repeatability and rationality of the work, we have added a detailed device preparation method flowchart and detailed testing conditions in Supplementary Note 1 and Supplementary Note 6
3. Added detailed performance characterization of the device, such as absorption spectra and carrier lifetime.

Author action: We have rewritten the various parts of the manuscript and highlighted them in red.

Comment 2: Most of the fundamental physics behind the device's working principle is only described qualitatively without sufficient modelling/simulation/references.

Reply 2: We appreciate the questions raised by the reviewer. We have systematically reviewed and sequentially described the physical processes governing the operation of the device. In this paper, the primary physical principles underlying this work can be mainly categorized into the electrical gain induced by fluctuating potential and the artificial anisotropic absorption resulting from specific optical grating structures. We applied Silvaco TCAD for electrical simulations and employed COMSOL for wave optics simulations to elucidate the device's operational mechanisms. In response to the questions, we have undertaken a detailed modeling and exploration of the device's operational mechanisms, and have made the following modifications:

1. Electrical gain analysis

The generation of electrical gain involves three main physical processes: the first is the generation of potential wells, the second is the separation of photocarriers, and the third is the process of photoconductive gain.

1.1 Generation of potential wells

It has been discovered that a strong electric field is located at the edge of silicon and silicon oxide, which would be stronger than the built-in electric field formed by graphene and silicon, which can be modeled and simulated in Figure R2.1 by Silvaco TCAD. For the slit structure, due to the electric field coupling between adjacent grating boundaries, the surface potential distribution will resemble a valley with a higher potential difference, as shown in the corresponding schematic diagram of Figure S5c. This potential distribution caused by the slit structure can be called the slit

effect. Furthermore, when a slit structure is constructed in two mutually perpendicular directions in a two-dimensional plane, the valley potential distribution will evolve into a 2D potential well.

Figure R2.1 Surface potential simulation. (a) 3D structural modeling. (b) Surface electric field distribution map. (c) The relationship between surface electric field and position.

As a verification, we measured the surface potential of the structure based on AFM characterization and obtained the electric field distribution maps that were consistent with the simulation, as shown in Figure R2.2.

Figure R2.2 Surface potential characterization. (a)-(b) AFM image of 2D slit structure and corresponding surface potential distribution characterization diagram. (c)-(d) 3D AFM image of slit structure and corresponding surface potential distribution characterization diagram

1.2 Separation of photocarriers

Figure R2.3. Schematic diagram of surface electric field distribution and mechanism of photo-generated carrier separation.

Based on the aforementioned fluctuation potential distribution, it significantly influences the separation of photocarriers. Since this work primarily focuses on the testing of the infrared spectrum, the generation of photocarriers relies on graphene. The separation of photocarriers mainly occurs through the action of the interface electric field between graphene and silicon as process ②, as well as the action of the transverse homojunction electric field of graphene for process ①, as shown in Figure R2.3.

The relationship between the energy band of the silicon/graphene interface and the position (z) is obtained by solving the Poisson equation: (Li, T. et al. *npj 2D Materials and Applications* 2 (2018).)

$$\varphi(z) = \begin{cases} \varphi(-w) & z < -w \\ \varphi(-w) - \frac{qN(z+w)^2}{2\epsilon_s} & -w < z < 0 \end{cases} \quad (\text{R1})$$

where $z=0$ is the interface between silicon and graphene, w is the width of depletion zone at the interface, $N = N_D - N_A$ is the net carrier concentration of silicon, where N_D is the donor impurity concentration, N_A is the acceptor impurity concentration, and ϵ_s is the dielectric constant of silicon.

The corresponding expression of the contact potential can be obtained by:

$$\psi_s^{G-S} = -q(\varphi(0) - \varphi(-w)) = \frac{qNw^2}{2\epsilon_s} \quad (\text{R2})$$

Since graphene will not have large Fermi level shift when contacting with silicon oxide, the electric field of homojunction ψ_s^{G-G} is equal to the edge strong electric field.

Next, under the action of electric potential ψ_s^{G-G} and ψ_s^{G-S} , the photocarriers in graphene are separated and form photocurrent. For convenience, we will refer to the two sides with potential differences as the P-type region and N-type region, respectively. For the carrier distribution in the heterojunction region, the relationship between carrier concentration of P-region and quasi Fermi level is

$$n_p = n_i \exp\left(\frac{E_{Fn} - E_i}{k_0 T}\right) \quad (R3)$$

$$p_p = n_i \exp\left(\frac{E_i - E_{Fp}}{k_0 T}\right) \quad (R4)$$

thus

$$n_p p_p = n_i^2 \exp\left(\frac{E_{Fn} - E_{Fp}}{k_0 T}\right) \quad (R5)$$

At the boundary of P region, we define $x = -x_p, E_{Fn} - E_{Fp} = qV$, so

$$n_p(-x_p) p_p(-x_p) = n_i^2 \exp\left(\frac{E_{Fn} - E_{Fp}}{k_0 T}\right) \quad (R6)$$

Because $p_p(-x_p)$ is the majority carrier in the P-type region, so $p_p(-x_p) = p_{p0}$, $n_{p0} p_{p0} = n_i^2$. Thus, at the boundary of P-type region $x = -x_p$, minority carrier concentration at P region is

$$n_p(-x_p) = n_{p0} \exp\left(\frac{qV}{k_0 T}\right) = n_{p0} \exp\left(\frac{qV - V_D}{k_0 T}\right) \quad (R7)$$

Thus, the photo-generated minority carrier concentration injected into the P-type region is obtained

$$\Delta n_p(-x_p) = n_p(-x_p) - n_{p0} = n_{p0} \left[\exp\left(\frac{qV}{k_0 T}\right) - 1 \right] \quad (R8)$$

Similarly, the photo-generated minority carrier concentration injected into N-type region at the boundary $x = -x_n$ is

$$\Delta p_n(x_n) = p_n(x_n) - p_{n0} = p_{n0} \left[\exp\left(\frac{qV}{k_0 T}\right) - 1 \right] \quad (R9)$$

It can be seen that the photo-generated minority carrier at the boundary of the injection barrier region is a function of the applied voltage and also a boundary condition for solving the continuity equation.

In the steady state, the continuity equation of photo-generated minority carriers in the hole diffusion region is

$$D_p \frac{d^2 \Delta p_n}{dx^2} - \mu_n \varepsilon_x \frac{d \Delta p_n}{dx} - \mu_n p_n \frac{d \varepsilon_x}{dx} - \frac{p_n - p_{n0}}{\tau_p} = 0 \quad (\text{R10})$$

In the case of small injection $\varepsilon_x = 0$

$$D_p \frac{d^2 \Delta p_n}{dx^2} - \frac{p_n - p_{n0}}{\tau_p} = 0 \quad (\text{R11})$$

The variation of carrier concentration in P and N regions can be solved:

$$\Delta p(x) = p_n(x) - p_{n0} = p_{n0} \left[\exp\left(\frac{qV}{k_0T}\right) - 1 \right] \exp\left(\frac{x_n - x}{L_p}\right) \quad (\text{R12})$$

$$\Delta n(x) = n_p(x) - n_{p0} = n_{p0} \left[\exp\left(\frac{qV}{k_0T}\right) - 1 \right] \exp\left(\frac{x_p + x}{L_n}\right) \quad (\text{R13})$$

where $qV = q(V_{bi} - V_{oc}) = \varphi_i$. $qV_{bi} = E_{Fn} - E_{Fp}$, qV_{bi} is contact potential difference or built-in potential difference due to the difference of Fermi energy levels of junctions in the dark state, where V_{oc} comes from the photogenerated voltage

$$V_{oc} = \frac{k_0T}{q} \ln\left(\frac{I_L}{I_s} - 1\right) \quad (\text{R14})$$

where Δn (Δp) is the injection concentration of electrons (holes), n_{p0} is the intrinsic carrier concentration, μ_n (μ_p) is the mobility of electrons (holes), E is the applied electric field. g is the generation rate of excess carriers. L is the length of graphene channel. qV_D is the barrier height due to the difference of Fermi energy levels of junctions in the dark state L_n (L_n) is the diffusion length of carrier.

It is also assumed that the holes in the diffusion length L_p and the electrons in L_n can diffuse to the other side of the pn junction. Then the photogenerated current is

$$I_L = q\bar{Q}A(L_p + L_n) \quad (\text{R15})$$

Where \bar{Q} is represented as the average generation rate of photo-generated carriers within the diffusion length ($L_p + L_n$) of the junction. The interface voltage of heterojunction can be derived

$$V = V_{bi} - \frac{k_0T}{q} \ln\left[\frac{q\bar{Q}A(L_p + L_n)}{I_s} - 1\right] \quad (\text{R16})$$

where I_s is the reverse saturation current, and $qV_{bi} \propto \psi_s^{G-S} + \psi_s^{G-G}$.

1.3 Photoconductive gain

Under the influence of the interface electric field, one type of photocarriers is

separated to the potential well region, while the other type of photocarriers, under the bias effect, undergo photoconduction within the graphene channel, resulting in photocurrent generation. The potential well traps photogenerated electrons/holes, thereby extending their recombination time. As a result, multiple conduction cycles occur within the channel before the recombination of photocarriers, leading to gain. We refer to this process as the photogating effect.

According to the Gain expression based on photogating effect (*Fang, H. & Hu, W. Advanced science 4, 1700323 (2017).*)

$$\begin{cases} I_{ph} = gALeG \\ I_{ph} = (\Delta\sigma \cdot E)A = (\Delta n\mu_n + \Delta p\mu_p)eEA \end{cases} \quad (R17)$$

The formed built-in electric field not only promotes the separation of photogenerated carriers in graphene, but also inhibits the recombination of photogenerated carriers, which can induce cyclic gain. After derivation with R16, the gain can be expressed as:

$$G = \frac{\Delta n(\mu_n + \mu_p)E}{gL} = \frac{n_{p0} \left[\exp\left(\frac{qV}{k_0T}\right) - 1 \right] \exp\left(\frac{x}{L_n}\right) (\mu_n + \mu_p)E}{gL} \quad (R18)$$

where $V = V_{bi} - \frac{k_0T}{q} \ln\left[\frac{q\bar{Q}A(L_p + L_n)}{I_s} - 1\right]$.

When there is no applied voltage on the junction, $G \propto V_{bi}$. Thus, the enhancement of the built-in potential can lead to the increase of the gain. Here, the separation driving force of photo-generated carriers is provided by the surface lateral electric field and the vertical heterojunction built-in electric field. Therefore, $qV_{bi} \propto \psi_s^{G-S} + \psi_s^{G-G}$.

2. Optical analysis

Throughout the entire process of photodetection, the first step involves the incidence and absorption of light, while the second step involves the conversion of light into an electrical signal (mentioned above). Therefore, for polarization-sensitive detection, this is achieved through the polarization ratio of incident and absorbed light. Specific-sized grating structures have the ability to induce different local and reflective properties for light polarized in different directions, which we refer to as artificial anisotropy. Here, silicon gratings exhibit different resonant effects on

polarized light parallel and perpendicular to the grating direction. Therefore, by adjusting the duty cycle of the silicon grating, polarization sensitive detection can be achieved by localizing polarized light of different wavelength. For example, during the process of adjusting the duty cycle from 0.3 to 0.4, the peak of the reflection spectrum exhibits a red shift, as shown in Figure R2.4. The polarized light reflection spectrum measured by FTIR is highly consistent with the simulation results. Near Field Distribution of DC=0.3 Grating structure at different polarization angles are shown in Figure R2.5.

Accordingly, by design, the device exhibits polarization-sensitive detection for polarized light with a wavelength of 4 μm when the duty cycle is 0.8, as shown in Figure R2.6.

Figure R2.4 (a)-(c) The polarization sensitive reflection spectra of structures with DC of 0.3, 0.35, and 0.4 measured by FTIR. The illustration shows the corresponding far-field simulation results. The grating height is 160 nm, the unit length is 1 μm

Figure R2.5 (a) Near field distribution of DC=0.3 Grating Structure at Different Polarization Angles (θ). $\lambda=1550$ nm. (b)-(c) Distribution of three-dimensional structures with θ of 90° and 0° entering the site, $\lambda=1550$ nm.

Figure R2.6 Absorption spectra of devices with different dielectric structures

Author action: We have updated the above descriptions on Supplementary Note 2 and Supplementary Note 3 respectively.

Comment 3. Figure 1d the simulation seems to show an asymmetric behavior in an otherwise (apparently)symmetric structure

Reply 3: We appreciate the questions raised by the reviewer. In Fig 1d. The left figure shows the three-dimensional distribution of the electric field of the 2D slit structure. It can be clearly seen that the strong electric field at the edge of the stripes can couple with the weak electric field at the center of the groove to form a valley potential. This distribution is also symmetrical in symmetric structures, as the image is a three-

dimensional structure with a certain angle from the top, so it does not appear clear enough. It can be seen from the cross-sectional view (figure right-bottom). This inhomogeneous staggered potential distribution will affect the separation and arrangement of photogenerated carriers in the plane.

Figure R2.7. Studies on the electric field distribution induced by slit structure.

Author Action: We have revised the three-dimensional potential distribution diagram in Figure 1 as below

Figure R2.8 Revised Figure R2.7.

Comment 4. Most of the figures and plots are not fully readable and in low-quality

Reply 4: I appreciate the comments made by the reviewer regarding the figures and plots. I have made the following improvements to each of the figures and plots:

1. Simulation and Layout of the Electric Field in Figure 1.
2. Adjust the aspect ratio of Figure 2 for aesthetic purposes.
3. Modified the labeling of Figure 3 for consistency.
4. Expanded the supplementary information to Figure S20.

Furthermore, improvements have been made to the arrangement of all figures and tables, as well as their corresponding descriptions.

Author Action: All the revisions concerning the figures and their descriptions throughout the entire manuscript have been updated in the manuscript and marked in red.

Comment 5. The responsivity mentioned is expressed without any further information on the characterization conditions: applied source-drain voltage, condition of the illumination, spot shape and size, etc. This info are also missing in the supplementary material

Reply 5: We appreciate the questions raised by the reviewer, and we apologize for not providing comprehensive information regarding the characterization of responsivity. These have been calculated and compiled during our testing, and the following are the actual test conditions and parameters, as shown in the table R2.1. All tests were conducted with a bias voltage of 0.1V.

Table R2.1 Parameters for Responsiveness Characterization

Wavelength (μm)	Spot area (mm^2)	Power (mW)	Optical power density (mW/cm^2)	Device area (mm^2)	Photocurrent (μA)	Responsivity (A/W)
1.55	38	10	26.32	0.0016	16	38
4	50	50	100	0.0016	10	6.25
6	28	100	357.14	0.0016	4	7
7	28	100	357.14	0.0016	3	5.25
8	28	100	357.14	0.0016	2	3.5
9	28	140	500	0.0016	3	3.75
10	28	130	464.29	0.0016	2.5	3.37
11	28	120	428.57	0.0016	1.5	2.19

The responsivity of device can be obtained by:

$$R = \frac{\Delta I}{P_e} = (I_{\text{light}} - I_{\text{dark}}) \times (P \times \frac{S_{\text{device}}}{S_{\text{light}}})^{-1} \quad (\text{S-1})$$

where ΔI is photocurrent, P is optical power, S_{device} and S_{light} are the areas of active region and spot, respectively. Here, optical power density can be defined as $\frac{P}{S_{\text{light}}}$.

Due to the fact that the spot area is significantly larger than the active area of the device, and the central region of the spot in which the device is located is far from the region of power decay at the edges, it can be approximated as a uniform distribution of optical power. Therefore, the definition of optical power density is more applicable here.

Author action: We have added the TableR2.1 as Table S4 of Supplementary Note 6 and added relevant instructions.

Comment 6. The caption of figure2 starts by saying "simulation...." But the overall picture does

not include mainly simulation.

Reply 6: We appreciate the questions raised by the reviewer. Figure 2 primarily describes the device performance and provides an electrical perspective analysis, in which Figure 2d shows the simulated schematic diagram of the manipulation of gate voltage on electric field distribution. The caption of Figure 2 can be revised as **Device performance and electrical test analysis**.

Author action: We have revised the caption of Figure 2 as **Device performance and electrical test analysis**.

Comment 7. Blackbody radiation measurements have several conceptual gaps. To mention a few: how are the other effects that can induce a variation in the graphene resistance taken into account (thermal, strain, etc)?

Reply 7: We sincerely thank the reviewer's comment on the Blackbody radiation measurements. We apologize for not providing sufficient information in the blackbody radiation section to facilitate the reader's understanding. Blackbody detection, as a detection standard for practical applications, is used to demonstrate the infrared detection performance of photodetectors. Here, our blackbody testing was conducted following previously reported methods as:

[1] Peng, M. et al. Room-Temperature Blackbody-Sensitive and Fast Infrared Photodetectors Based on 2D Tellurium/Graphene Van der Waals Heterojunction. *ACS Photonics* **9**, 1775-1782 (2022).

[2] Wang, Y. et al. Fast Uncooled Mid-Wavelength Infrared Photodetectors with Heterostructures of van der Waals on Epitaxial HgCdTe. *Advanced materials* **34**, e2107772 (2022).

[3] Peng, M. et al. Blackbody-sensitive room-temperature infrared photodetectors based on low-dimensional tellurium grown by chemical vapor deposition. *Science advances* **7**, eabf7358 (2021).

The blackbody source is Cisystems SR200N with an adjustable temperature from 500K to 1200K. The device was placed in front of the aperture with a fixed modulation frequency of chopped by an optical chopper wheel. The blackbody is more like a simulated heat source, with its radiation power significantly lower than the power provided by the laser, resulting in minimal thermal fluctuations in graphene. As the reviewer pointed out, given graphene's thermoelectric effect, we can demonstrate this by testing how the resistance of graphene varies with blackbody temperature, as shown in Figure R2.9. The changes in blackbody temperature do not result in significant thermal fluctuations in graphene

Figure R2.9 The relationship between the resistance of devices with different configuration structures as a function of blackbody temperature.

Regarding the impact of graphene stress on device performance, it is undeniable that a structured surface with variations can introduce stress to graphene. However, the introduction of stress in graphene is primarily due to the stretching and folding induced by the structure. In this paper, the height and unit length of the structure are consistent, resulting in approximately similar levels of strain. The variations in device performance are likely more attributable to the influence of the interfacial electric field. Therefore, strain should not be considered as a variable causing systematic variations in the performance of the device in this study.

Author Action: Related description “The blackbody source is Cisystems SR200N with an adjustable temperature from 500K to 1200K. The device was placed in front of the aperture with a fixed modulation frequency of chopped by an optical chopper wheel.” and “During the testing process, there was no significant change in the device's resistance, thus the interference caused by photothermal effects can be neglected.” Have been added on page 12 of manuscript.

Comment 8. Supplementary material: the “Surface Electric Field Engineering and Gain model” is mainly a standard model for pn junction in and out of equilibrium. The authors should explicitly adapt the model to the actual device under test.

Reply 8: Thank you for the questions raised by the reviewers. We have reorganized and reformulated the physical processes in the electrical section of our work. The generation of electrical gain involves three main physical processes: the first is the generation of potential wells, the second is the separation of photocarriers, and the third is the process of photoconductive gain.

For the the separation of photocarriers, under the action of electric potential ψ_s^{G-G} and ψ_s^{G-S} mentioned above, the photocarriers in graphene are separated and form photocurrent. For convenience, we will refer to the two sides with potential differences as the P-type region and N-type region, respectively. For the carrier distribution in the heterojunction region, the relationship between carrier concentration of P-region and quasi Fermi level is

$$n_p = n_i \exp\left(\frac{E_{Fn} - E_i}{k_0 T}\right) \quad (R3)$$

$$p_p = n_i \exp\left(\frac{E_i - E_{Fp}}{k_0 T}\right) \quad (R4)$$

thus

$$n_p p_p = n_i^2 \exp\left(\frac{E_{Fn} - E_{Fp}}{k_0 T}\right) \quad (R5)$$

At the boundary of P region, we define $x = -x_p, E_{Fn} - E_{Fp} = qV$, so

$$n_p(-x_p) p_p(-x_p) = n_i^2 \exp\left(\frac{E_{Fn} - E_{Fp}}{k_0 T}\right) \quad (R6)$$

Because $p_p(-x_p)$ is the majority carrier in the P-type region, so $p_p(-x_p) = p_{p0}$, $n_{p0} p_{p0} = n_i^2$, Thus, at the boundary of P-type region $x = -x_p$, minority carrier concentration at P region is

$$n_p(-x_p) = n_{p0} \exp\left(\frac{qV}{k_0 T}\right) = n_{p0} \exp\left(\frac{qV - V_D}{k_0 T}\right) \quad (R7)$$

Thus, the photo-generated minority carrier concentration injected into the P-type region is obtained

$$\Delta n_p(-x_p) = n_p(-x_p) - n_{p0} = n_{p0} \left[\exp\left(\frac{qV}{k_0 T}\right) - 1 \right] \quad (R8)$$

Similarly, the photo-generated minority carrier concentration injected into N-type region at the boundary $x = -x_n$ is

$$\Delta p_n(x_n) = p_n(x_n) - p_{n0} = p_{n0} \left[\exp\left(\frac{qV}{k_0 T}\right) - 1 \right] \quad (R9)$$

It can be seen that the photo-generated minority carrier at the boundary of the injection barrier region is a function of the applied voltage and also a boundary condition for solving the continuity equation.

In the steady state, the continuity equation of photo-generated minority carriers in the hole diffusion region is

$$D_p \frac{d^2 \Delta p_n}{dx^2} - \mu_n \varepsilon_x \frac{d \Delta p_n}{dx} - \mu_n p_n \frac{d \varepsilon_x}{dx} - \frac{p_n - p_{n0}}{\tau_p} = 0 \quad (\text{R10})$$

In the case of small injection $\varepsilon_x = 0$

$$D_p \frac{d^2 \Delta p_n}{dx^2} - \frac{p_n - p_{n0}}{\tau_p} = 0 \quad (\text{R11})$$

The variation of carrier concentration in P and N regions can be solved:

$$\Delta p(x) = p_n(x) - p_{n0} = p_{n0} \left[\exp\left(\frac{qV}{k_0T}\right) - 1 \right] \exp\left(\frac{x_n - x}{L_p}\right) \quad (\text{R12})$$

$$\Delta n(x) = n_p(x) - n_{p0} = n_{p0} \left[\exp\left(\frac{qV}{k_0T}\right) - 1 \right] \exp\left(\frac{x_p + x}{L_n}\right) \quad (\text{R13})$$

where $qV = q(V_{bi} - V_{oc}) = \varphi_i$. $qV_{bi} = E_{Fn} - E_{Fp}$, qV_{bi} is contact potential difference or built-in potential difference due to the difference of Fermi energy levels of junctions in the dark state, where V_{oc} comes from the photogenerated voltage

$$V_{oc} = \frac{k_0T}{q} \ln\left(\frac{I_L}{I_s} - 1\right) \quad (\text{R14})$$

where Δn (Δp) is the injection concentration of electrons (holes), n_{p0} is the intrinsic carrier concentration, μ_n (μ_p) is the mobility of electrons (holes), E is the applied electric field. g is the generation rate of excess carriers. L is the length of graphene channel. qV_D is the barrier height due to the difference of Fermi energy levels of junctions in the dark state L_n (L_n) is the diffusion length of carrier.

It is also assumed that the holes in the diffusion length L_p and the electrons in L_n can diffuse to the other side of the pn junction. Then the photogenerated current is

$$I_L = q\bar{Q}A(L_p + L_n) \quad (\text{R15})$$

Where \bar{Q} is represented as the average generation rate of photo-generated carriers within the diffusion length ($L_p + L_n$) of the junction. The interface voltage of heterojunction can be derived

$$V = V_{bi} - \frac{k_0T}{q} \ln\left[\frac{q\bar{Q}A(L_p + L_n)}{I_s} - 1\right] \quad (\text{R16})$$

where I_s is the reverse saturation current, and $qV_{bi} \propto \psi_s^{G-S} + \psi_s^{G-G}$.

Author Action: We have added relevant expressions to Supplementary Note 2

Comment 9. No measurements of the graphene have been carried out. Raman, doping, strain and resistivity should be carried out. If some of them are not possible, at least proper characterization of the resistivity should be done.

Reply 9: We appreciate the questions raised by the reviewer, and we have conducted

detailed parameter characterization for graphene as follows, including Parameters of the graphene measured by Hall effect testing, I-V curves of devices with different structures and Raman testing of graphene on silicon oxide:

Table R2.2 Parameters of the graphene measured by Hall effect testing

Parameter	Mobility (cm ² /V·s)	Concentration(/cm ²)	Type of doping	Resistance (Ω)
Quantity	8116.2	7.8×10 ¹¹	P	1135

Figure R2.10. (a) I-V curves of devices with different structures. (b) Raman testing of graphene on silicon oxide.

Author Action: We have updated the above figures and tables in Supplementary Note 1 of the supplementary information.

Comment 10. Supplementary information: section "Transient photoresponses of graphene/2D slit structure photodetector upon different wavelength". The chopper speed is properly set to show the transient. Hence this can't be properly observed/quantified.

Reply 10: We appreciate the questions raised by the reviewer. The section "Transient photoresponses of graphene/2D slit structure photodetector upon different wavelength" refers to the I-time light response characteristics of the device when the optical signal is modulated with an 8-second periodic switching signal. The Figure caption is more appropriately corrected to 'Photoswitching behaviors of graphene/2D slit structure photodetector upon different wavelength'

Author Action: We have revised the caption of Figure S8 as "Photoswitching behaviors of graphene/2D slit structure photodetector upon different wavelength".

Comment 11. Supplementary material: in Table S2, several working principles are compared together: this comparison requires supporting evidence and explanations in order to give the tools

to the reader to understand the quantities.

Reply 11: We appreciate the questions raised by the reviewer. The device type in this paper is a photoconductive device based on the photogating effect, distinct from photodiodes and photo-thermoelectric devices, each of these three device types has its own advantages and disadvantages. Photodiodes and photo-thermoelectric devices utilize the separation and flow of photocarriers under asymmetric potential or temperature gradients, allowing them to operate at zero bias voltage, with low dark current and a high ON/OFF ratio. However, they lack gain and exhibit low responsivity as disadvantages. Photoconductive devices based on photogating effect primarily rely on the separation and recombination of charge carriers in vertical heterojunctions or traps to generate gain, necessitating operation under bias voltage, hence achieving high responsivity.

As the reviewer has pointed out, different types of devices need to be discussed and compared separately, as shown in the table below. All of these devices rely on graphene for light absorption, and, therefore, for broad-spectrum detection, photoconductive devices with gain prove to be advantageous. Our work leverages the synergistic interaction of the optical and electrical fields to break through the performance advantages reported in existing literature.

Table R2.3 Summary of device parameters of several typical graphene/semiconductor photodetectors previously reported, and our own device.

Types of devices	Working mechanism	Responsivity	Wavelength
Graphene/Si ⁸	Photodiode	0.28 A/W	1550 nm
TPA-doped tri-layer graphene/Si ⁹	Photodiode	0.435 A/W	800 nm
Graphene/CNT/SiO ₂ /Si ¹²	Photodiode	0.21 A/W	980 nm
Graphene–silicon-on-insulator ¹³	Photodiode	0.029 A/W	980 nm
Graphene double-layer heterostructure ¹⁶	photo-thermoelectric	1.1 A/W	3200 nm
Graphene–silicon heterojunction in conductor mode ¹⁰	Photoconductor	0.23 A/W	1550 nm
PtNPs/graphene/Si ¹¹	Photoconductor	26 A/W	790 nm
MoTe ₂ /Graphene ¹⁵	Photoconductor	60 A/W	1064 nm
Au NP array/graphene ¹⁷	Photoconductor	83 A/W	1550nm

Graphene/ WS ₂ ¹⁸	Photoconductor	0.735 A/W	1550nm
Graphene/silicon grating ¹⁹	Photoconductor	25 A/W	2700nm
This work	Photogating	0.2 A/W-38 A/W	1500-11000 nm

Author Action: We have added the TableR2.3 as Table S4 of Supplementary Note 6 and added relevant instructions.

Reviewer 3

Comments:

This manuscript titled “Synergistic-Potential Engineering Enables High-Efficiency Graphene Photodetector for Near to Mid-infrared light” by Jiang et al reported the design and fabrication of graphene infrared photodetector based on 2D silicon-on-insulator substrate. The 2D potential well created by the patterned silicon block matrix can effectively trap photoexcited carriers and enable high photoconductive gain. In addition, by designing the silicon matrix for polarized detection, the detector showed highly polarized photo response for 1.55 μm incident light. Finally, the detector was tested for detecting blackbody radiation from 500 K to 1000 K with high responsivity and detectivity. This manuscript provided a new design for enhanced infrared photodetection of graphene photodetector. The patterned silicon substrate is compatible with semiconductor processing technology, and the design can be tailored for various bands. Therefore, this manuscript is of great important for graphene infrared photodetectors. I would recommend its publication after the following comments are properly addressed in the revised manuscript.

Author Reply: We appreciate the reviewer's positive feedback on this work and the valuable suggestions provided. We have made revisions and additions as per each point, including the inclusion of necessary figures and tables, as detailed below.

1. The trapping of carriers by the potential well is crucial for the high responsivity, and the recombination lifetime is an important parameter. Can the authors measure or estimate the prolonged the carrier lifetime compared with conventional graphene/Si junction?

Author Reply: We appreciate the questions raised by the reviewer. The lifetime-gain factor is mainly determined by the recombination lifetime τ_r and the mobility of conduction carriers μ . In this work, the improvement in responsivity stems from interface field engineering, which simultaneously enhances lifetime gain and potential gain. The recombination lifetime of photogenerated charge carriers can be obtained by studying the falling edge of the device's switching response under 1000 K blackbody radiation, as shown in Figure R3.1. The two-dimensional slit structure device with the highest responsivity also exhibits a longer recombination lifetime.

Figure R3.1 Falling edge time of devices with different dielectric structures under 1000 K blackbody radiation.

Author action: We have supplemented Figure R3.1 as Figure S19 and added description “Where the estimation of device lifetime can be seen in the test of Figure S19.” on page 7 of manuscript.

2. The detector showed responsivity up to 38 A/W. Is this high responsivity mainly from the long carrier lifetime or enhanced absorption? Did the authors measure the absorption of the device?

Author Reply: We appreciate the questions raised by the reviewer. The devices in this paper rely on the synergistic interaction of the optical and electric fields, where the high responsivity primarily arises from waveband-immunity electrical gain induced by the slit structure. Strong polarization-sensitive absorption due to the dielectric grating is only prominent in the resonant band.

Based on the absorption spectra obtained from tests of devices with different structures as Figure R3.2, resonant bands at DC = 0.3 and 0.8 are observed near 1.55 μm and 4 μm , respectively, and they are sensitive to linear polarization. In the case of the two-dimensional slit structure, it did not lead to enhanced absorption, and its high gain primarily arises from the electric field gain.

Figure R3.2 Absorption spectra of devices with different dielectric structures

Author action: We have supplemented Figure R3.1 as Figure S12.

3. To achieve polarized detection, the silicon needs to be patterned for polarization sensitive response, however, the responsivity is much lower than that shown in Figure 2. It seems the 2D potential well and the polarized detection can not be attained simultaneously. Please comment on this.

Author Reply: We greatly appreciate the valuable questions raised by the reviewer. Periodic grating structures introduce non-uniform interface electric fields, thus enhancing the device's quantum efficiency and gain. However, in the case of configurable dielectric structures, the two-dimensional slit structure exhibits the most significant gain enhancement, thereby achieving the highest responsivity. For other periodic dielectric structures, the DC = 0.3 and DC = 0.8 structures enhance responsivity while being sensitive to polarization at 1.55 μm and 4 μm, respectively, enabling high responsivity polarization-sensitive detection through the synergistic interaction of the optical and electric fields. The responsivities in Figures 2 and 3 are corresponding and consistent. However, for the two-dimensional slit structure, there is no specific resonance mode for the test bands, and it can only be used to highlight its high responsivity characteristics.

Author action: We have added the relevant description “Nevertheless, in the case of the 2D slit structure, no particular resonance mode exists for the test bands, making it suitable for emphasizing its remarkably high responsivity characteristics.” on page 10 of the manuscript.

4. For blackbody detection, the responsivity increases with temperature. Since the radiation power also increases with temperature, this trend is opposite to the normally observed decreasing responsivity with incident power. Did the author measured power dependent responsivity for 1.55 μm or other wavelength?

Author Reply: We appreciate the questions raised by the reviewer. In the device's laser testing, there is an attenuation relationship between the device's responsivity and optical power, which does not apply to blackbody testing. First, according to the radiation rule for blackbody testing (Figure R3.3 a), the variation in optical power with blackbody temperature has a small span, limited to within one order of magnitude, which is insufficient to demonstrate a decay relationship. Meanwhile, unlike laser testing, with an increase in temperature, the radiation peak shifts towards shorter wavelengths (Figure R3.3 b), and the device's quantum efficiency also improves. Therefore, the responsivity experiences a slight increase. Here, our blackbody testing was conducted following previously reported methods to ensure the validity of the tests, the observed patterns align with those reported in the literature as:

[1] Peng, M. et al. Room-Temperature Blackbody-Sensitive and Fast Infrared Photodetectors Based on 2D Tellurium/Graphene Van der Waals Heterojunction. *ACS Photonics* **9**, 1775-1782 (2022).

[2] Wang, Y. et al. Fast Uncooled Mid-Wavelength Infrared Photodetectors with Heterostructures of van der Waals on Epitaxial HgCdTe. *Advanced materials* **34**, e2107772 (2022).

[3] Peng, M. et al. Blackbody-sensitive room-temperature infrared photodetectors based on low-dimensional tellurium grown by chemical vapor deposition. *Science advances* **7**, eabf7358 (2021).

Figure R3.3 (a) The relationship between the real radiation power and the blackbody temperature considering factors of radiation distance, spot size, device size, and background radiation. (b) Blackbody measurement schematic of graphene/nanostructured all-dielectric photodetectors under blackbody source illumination.

Figure R3.4 The relationship between device responsivity and 1.55 μm laser power density

Additionally, we have added the relationship between device responsivity and 1.55 μm laser power as Figure R3.4, as mentioned by the reviewer, which exhibits a linear decay relationship on a logarithmic scale.

Author action: We have added the Figure R3.4 as Figure S10 in Support Information.

5. Table S2 is not complete.

Author Reply: We appreciate the reviewer's reminder, and we have improved and categorized the Table S2 while adding a systematic discussion of devices with different working mechanisms as Table R3.1.

Table R3.1 Summary of device parameters of several typical graphene/semiconductor photodetectors previously reported, and our own device.

Types of devices	Working mechanism	Responsivity	Wavelength
Graphene/Si ⁸	Photodiode	0.28 A/W	1550 nm
TPA-doped tri-layer graphene/Si ⁹	Photodiode	0.435 A/W	800 nm
Graphene/CNT/SiO ₂ /Si ¹²	Photodiode	0.21 A/W	980 nm
Graphene–silicon-on-insulator ¹³	Photodiode	0.029 A/W	980 nm
Graphene double-layer heterostructure ¹⁶	photo-thermoelectric	1.1 A/W	3200 nm
Graphene–silicon heterojunction in conductor mode ¹⁰	Photoconductor	0.23 A/W	1550 nm
PtNPs/graphene/Si ¹¹	Photoconductor	26 A/W	790 nm
MoTe ₂ /Graphene ¹⁵	Photoconductor	60 A/W	1064 nm
Au NP array/graphene ¹⁷	Photoconductor	83 A/W	1550nm

Graphene/ WS ₂ ¹⁸	Photoconductor	0.735 A/W	1550nm
Graphene/silicon grating ¹⁹	Photoconductor	25 A/W	2700nm
This work	Photogating	0.2 A/W-38 A/W	1500-11000 nm

Author action: We have added Table R3.1 as Table S4 in Support Information.

6. *What was the source-drain voltage and the corresponding electric field?*

Author Reply: We appreciate the questions raised by the reviewer, and we apologize for not providing sufficient explanations. All device testing and electric field simulations in the paper were conducted at a bias voltage of 0.1 V.

Author action: We have added bias voltage conditions in Figures 2, 3, and 4 of the manuscript, respectively.

REVIEWERS' COMMENTS

Reviewer #1 (Remarks to the Author):

The authors have addressed all my concerns and their work is, in my opinion, ready to be published.

Reviewer #2 (Remarks to the Author):

The manuscript presents an interesting implementation of a layered material-based device where the electrical gain and light field tuning effects are combined in order to increase the responsivity and to extend the working range toward the long(er) wavelength.

The authors provided satisfactory answers to the raised comments and the reviewed version can be recommended for publication.

Reviewer #3 (Remarks to the Author):

The authors have properly addressed most reviewers' concerns, and the revised manuscript has been improved substantially. I would recommend its publication.